# Zero-shot Learning via Simultaneous Generating and Learning

**Hyeonwoo Yu**   **Beomhee Lee**
Automation and Systems Research Institute (ASRI)
Dept. of Electrical and Computer Engineering
Seoul National University
{bgus2000,bhlee}@snu.ac.kr

## Abstract

To overcome the absence of training data for unseen classes, conventional zero-shot learning approaches mainly train their model on seen datapoints and leverage the semantic descriptions for both seen and unseen classes. Beyond exploiting relations between classes of seen and unseen, we present a deep generative model to provide the model with experience about both seen and unseen classes. Based on the variational auto-encoder with class-specific multi-modal prior, the proposed method learns the conditional distribution of seen and unseen classes. In order to circumvent the need for samples of unseen classes, we treat the non-existing data as missing examples. That is, our network aims to find optimal unseen datapoints and model parameters, by iteratively following the generating and learning strategy. Since we obtain the conditional generative model for both seen and unseen classes, classification as well as generation can be performed directly without any off-the-shell classifiers. In experimental results, we demonstrate that the proposed generating and learning strategy makes the model achieve the outperforming results compared to that trained only on the seen classes, and also to the several state-of-the-art methods.

## 1   Introduction

The combination of the large amount of data and deep learning finds the usage in various fields such as machine learning and artificial intelligence. However, deep learning as a non-linear regression tool based on statistics mostly suffers from the insufficient or non-existing training data, which is the usual case and should be overcome for autonomous learning systems. The advantage of deep learning, that learns reliable models on plenty of labeled training datapoints, becomes a curse in this scenario, since the model loses the generalization aspects with lack of training data. This severely interrupts the scalability to unseen classes of which training samples simply does not exist.

Zero-shot learning (ZSL) is a learning paradigm that proposes an elegant way to fulfill this desideratum by utilizing semantic descriptions of seen and unseen classes [8, 30]. These descriptions are usually assumed to be given as the form of the class embedding vectors or textual descriptions of each class. By assuming that seen and unseen classes share the same class attribute space, transferring the knowledge from the seen to unseen can be achieved by training models on seen samples and plugging in embedding vectors of unseen classes. Based on this concept, previous works find a relation between class embedding vectors and given datapoints of classes, by learning a projection from feature vectors to the class attribute space [16, 22, 19]. Similar works can be conducted by learning a visual-semantic mapping using either shallow or deep embeddings, thereby handle the unseen datapoints via an indirect manner [35, 36, 17, 21, 3, 34]. These approaches have shown promising results. However,

intra-class variation is hardly considered which is inevitable to catch the more realistic situations, since the methods assume that each class is represented as a deterministic vector.

Thanks to the advents of deep generative model, which enable us to unravel the data in complex structure, one can overcome the scarce of unseen examples by directly generating samples from learned distribution. With the generated datapoints, ZSL can be viewed as a traditional classification problem. This scenario thus becomes an excellent testbed for evaluating the generalization of generative models [29], and several approaches are presented to directly generate datapoints for unseen classes by exploiting semantic descriptions [18, 27, 29, 28, 15, 37]. Under the assumption that the model which generates the high-quality samples for seen classes is also expected to have the similar results on unseen classes, these approaches mainly train conditional generative models on seen samples and plug the unseen class attribute vectors into their model to generate unseen samples. They subsequently train off-the-shell classifier such as SVM or softmax. However, as far as it goes, the proposed models are trained mainly on the seen classes. Obtaining the generative model for both seen and unseen is quite far from their consideration, since scarcity of the unseen samples is apparently a fundamental problem for ZSL.

We therefore propose a training strategy to obtain a generative model which experiences both seen and unseen classes. We treat unseen datapoints as missing data, and also variables that can be optimized like model parameters. Optimal model parameter requires the optimal training data, and optimal unseen samples can be sampled from the distribution expressed with optimal model parameters. To relieve this chicken-egg problem, we lean to the Expectation-Maximization (EM) method, which enables the model to Simultaneously be Generating And Learning (SGAL). That is, while training, we iteratively generate samples from current model and update networks based on that currently generated samples. For our model, a variational auto-encoder (VAE) [12] with category-specific multi-modal prior is leveraged. Since we aim to have a multi-modal VAE (mmVAE) that covers both seen and unseen classes, no additional classifier is needed and the encoder can directly serve as a classifier. In our case, model uncertainty can be an obstacle while generating samples and training model, since the model does not see the real unseen datapoints, and estimated samples for training are generated from the model. We thus exploit dropout which makes model take into account the distributions of model parameters [9], and neutralize the model uncertainty by activating dropouts when sampling estimated datapoints during training procedures.

## 2 Related Work

### 2.1 Conditional VAE and Category Clustering in Latent Space

In order to exploit the labeled dataset for generative model, several methods based on VAE are introduced. By modifying the Bayesian graph model of vanilla VAE, [23] and [13] utilizes labels of datapoints as the input of both encoder and decoder. Since they mainly focus on conditionally generating datapoints from trained model especially with the decoder, they assume the prior as isotropic Gaussian to simplify the formulation and network structure.

Several methods utilize the latent space with explicitly structured prior distributions. Beyond a fixed Gaussian prior suffering from little variability, [32, 33, 27, 7, 13] use gaussian mixture model (GMM) prior, whose multi-modal is set to catch multiple types of categories. Especially [7] proposes unsupervised clustering method with this latent prior, by learning multi-modal prior and VAE together. To categorize the training data with conditional generative model, [33] and [28] exploit a category-specific multi-modal prior. With distinct clusters according to the category or instance, they perform classifications using trained encoder as feature extractor. In addition, [33] further uses the model as an observation model for data association rather than classifier only, and presents the applications for probabilistic semantic SLAM.

### 2.2 Zero-shot Learning and Generative Model

ZSL possesses a challenging setting that the training and test dataset are disjoint in category context, thus traditional non-linear regression is hardly applied. Therefore, several indirect methods are proposed. [16] handles the problem by solving related sub-problems. [19, 35, 6] exploit the relations between each class, and express unseen classes as a mixture of proportions of seen classes. [1, 8, 21, 22, 24, 3] train their model to find a similarity between datapoints and classes.

In order to overcome the scarceness of unseen samples directly, conditional generative models in variations are exploited. [18, 15] exploit conditional VAE (CVAE) to generate conditional samples; [15] adds several regressor and restrictor, to let the model be more robust when generating unseen datapoints. [28] proposes a method based on VAE, with category-specific prior distribution. Generative adversarial network (GAN) is also exploited and shows promising results since sharpness and realism of generated samples are high enough [29]. Commonly, these methods based on deep generative model train their models first and generate enough samples for unseen classes and subsequently train additional classifier, rather than training conditional generative models for both seen and unseen classes. We thus present a deep generative model for both seen and unseen, which enables us to use the model as a classifier as well as a generator. Our model is single VAE, and end-to-end training is possible without training additional off-the-shell classifier.

## 3 Proposed Method

### 3.1 Problem Scenario

Suppose we have some dataset $\{\mathcal{X}^{s*}, \mathcal{Y}^{s*}\}$ of $S$ seen classes; a set of datapoints $\mathcal{X}^{s*} = \{\boldsymbol{x}_i^{s*}\}_{i=1}^{N^s}$ and their corresponding labels $\mathcal{Y}^{s*} = \{y_i^{s*}\}_{i=1}^{N^s}$, which are sampled from the true distribution $p\left(\boldsymbol{x}^s|y^s\right)$. $N^s$ is the number of sampled datapoints, and $y^{s*} \in \boldsymbol{L}^s = \{1, ..., S\}$. In the ZSL problem, we aim to have a model which can classify the datapoints of unseen classes $\mathcal{X}^{u*} = \{\boldsymbol{x}_j^{u*}\}_{j=1}^{N^u}$ labeled as $\mathcal{Y}^{u*} = \{y_j^{u*}\}_{j=1}^{N^u}$, where $y^{u*} \in \boldsymbol{L}^u = \{S+1, ..., S+U\}$. Clearly, $\boldsymbol{L}^s \cap \boldsymbol{L}^u = \varnothing$ and at training we have no corresponding datapoints for unseen classes. Yet in surrogate we have class semantic embedding (or class attribute) vectors $\boldsymbol{A}^* = \{\boldsymbol{a}_k^*\}_{k=1}^{S+U}$ for both seen and unseen classes, that describe the corresponding class and further imply the relations between classes. Note that each class has a distinct attribute vector, for example of seen classes $\boldsymbol{A}^{s*} = \{\boldsymbol{a}_k^*\}_{k=1}^{S}$, and we can express the corresponding classes of $\mathcal{X}^{s*}$ with attribute vectors as $\boldsymbol{A}_y^{s*} = \{\boldsymbol{a}_{y_i^{s*}}^*\}_{i=1}^{N^s}$.

### 3.2 Category-Specific Multi-Modal Prior and Classification

In order to capture the complex distribution, VAE can be a useful tool. Especially with labeled datapoints, CVAE can be utilized which approximates the conditional likelihood $p\left(\boldsymbol{x}|y\right)$ with the following lower bound [23]:

$$\mathcal{L}\left(\theta, \phi; \boldsymbol{x}, y\right) = -KL\left(q_\phi\left(\boldsymbol{z}|\boldsymbol{x}, y\right) || p\left(\boldsymbol{z}|y\right)\right) + \mathbb{E}_{\boldsymbol{z} \sim q}\left[\log p_\theta\left(\boldsymbol{x}|\boldsymbol{z}, y\right)\right]. \qquad (1)$$

However, since this model is designed to generate samples having certain desired properties such as category $y$, encoder $q_\phi\left(\boldsymbol{z}|\boldsymbol{x}, y\right)$ and decoder $p_\theta\left(\boldsymbol{x}|\boldsymbol{z}, y\right)$ need $y$ for both training and testing. Hence, for the classification task performed with datapoints of which labels are missing, both encoder and decoder are hardly exploited and decoder only takes advantage when generating datapoints in certain condition. Often, to relax the conditional constraint, the latent prior $p\left(\boldsymbol{z}|y\right)$ in (1) is assumed as $p\left(\boldsymbol{z}\right)$ which is independent to input variables; exploiting the latent variables for classification becomes another challenge.

We therefore assume that categories represented as class embedding vectors $\boldsymbol{a}^*$ cast a Bayesian dice via latent variable $\boldsymbol{z}$ to generate $\boldsymbol{x}$. For $\mathcal{X}^{s*}$ and $\boldsymbol{A}_y^{s*}$, the total marginal likelihood comprises a sum over that of individual datapoints $\log p\left(\mathcal{X}^{s*}|\boldsymbol{A}_y^{s*}\right) = \sum_i \log p\left(\boldsymbol{x}_i^{s*}|\boldsymbol{a}_{y_i^{s*}}^*\right)$, we then have:

$$\mathcal{L}\left(\Theta; \boldsymbol{x}^{s*}, \boldsymbol{a}_{y_i^{s*}}^*\right) = -KL\left(q_\phi\left(\boldsymbol{z}|\boldsymbol{x}^{s*}\right) || p_\psi\left(\boldsymbol{z}|\boldsymbol{a}_{y_i^{s*}}^*\right)\right) + \mathbb{E}_{\boldsymbol{z} \sim q}\left[\log p_\theta\left(\boldsymbol{x}^{s*}|\boldsymbol{z}\right)\right], \qquad (2)$$

where $\Theta = (\theta, \phi, \psi)$. In contrast to the traditional VAE, since our purpose is classification, we assume the conditional prior to be a category-specific Gaussian distribution [27, 28, 32, 33]. Then the prior can be expressed as $p\left(\boldsymbol{z}\right) = \sum_i p\left(\boldsymbol{a}_{y_i^{s*}}^*\right) p_\psi\left(\boldsymbol{z}|\boldsymbol{a}_{y_i^{s*}}^*\right)$ which is a multi-modal, and $p_\psi\left(\boldsymbol{z}|\boldsymbol{a}\right) = \mathcal{N}\left(\boldsymbol{z}; \boldsymbol{\mu}\left(\boldsymbol{a}\right), \boldsymbol{\Sigma}\left(\boldsymbol{a}\right)\right)$ where $\left(\boldsymbol{\mu}\left(\boldsymbol{a}\right), \boldsymbol{\Sigma}\left(\boldsymbol{a}\right)\right) = f_\psi\left(\boldsymbol{a}\right)$, which is a non-linear function implemented by, namely, prior network. In order to make the conditional prior simple and distinct according to categories, we follow the basic settings of [32, 33]; we simply let $\boldsymbol{\Sigma}\left(\boldsymbol{a}\right) = \boldsymbol{I}$, and adopt the prior regularization loss which promotes each cluster of $p_\psi\left(\boldsymbol{z}|\boldsymbol{a}\right)$ to be far away from all other clusters above the certain distance in latent space. The KL-divergence in (2) encourages the

variational likelihood $q_\phi$ to be overlapped on the corresponding conditional prior distinct according to the categories, thus encoded features are naturally clustered [28]. Since (2) approximates the true conditional likelihood, Maximum Likelihood Estimation (MLE) of optimal label $\hat{y}$ can be formulated as the following [32]:

$$\hat{y} = \underset{y^{s*}}{\operatorname{argmax}}\, p\left(\boldsymbol{x}^{s*}|\boldsymbol{a}_{y^{s*}}^*\right) \simeq \underset{y^{s*}}{\operatorname{argmax}}\, p_\psi\left(\boldsymbol{z} = \boldsymbol{\mu}\left(\boldsymbol{x}^{s*}\right)|\boldsymbol{a}_{y^{s*}}^*\right), \qquad (3)$$

where $\boldsymbol{\mu}\left(\boldsymbol{x}^{s*}\right)$ is the mean of the approximated variational likelihood $q_\phi\left(\boldsymbol{z}|\boldsymbol{x}^{s*}\right)$. By simply calculating Euclidian distances between category-specific multi-modal and the encoded variable $\boldsymbol{\mu}\left(\boldsymbol{x}^{s*}\right)$, classification results can be achieved. In other words, as shown in Fig. 2(a), encoded features and conditional priors can be easily utilized for classification, rather than simply abandon the encoder after training. The optimal parameter $\hat{\Theta}$ can be obtained by maximizing the lower bound in (2), which are for the datapoints of seen classes. Note that when training is converged, the conditional priors and variational likelihoods of unseen classes can be obtained by plugging in their associated class embedding vectors $\boldsymbol{A}^{u*} = \{\boldsymbol{a}_k^*\}_{k=S+1}^{S+U}$. In this way, we can perform classification task for both seen and unseen classes with (3), or generate datapoints for unseen classes by sampling from $p\left(\boldsymbol{x}|\boldsymbol{a}_y^{u*}\right) \simeq \int_{\boldsymbol{z}} p_{\hat{\theta}}\left(\boldsymbol{x}|\boldsymbol{z}\right) p_{\hat{\psi}}\left(\boldsymbol{z}|\boldsymbol{a}_y^{u*}\right) d\boldsymbol{z}$, and train additional classifier similar to [18].

### 3.3 Generative Model for both Seen and Unseen Classes

Even the model is trained on seen classes $\boldsymbol{A}^{s*}$, we can try to use the generative model by simply inputting the embedding vector of unseen classes $\boldsymbol{A}^{u*}$. However, the optimal parameters $\hat{\Theta}$ obtained by maximizing (2) are still fitted to the datapoints of seen classes, and hardly guarantee the exact regression results for unseen classes. In other words, the model represented by parameters has in effect no experience with unseen classes. To approximate the distribution for both seen and unseen classes, certainly it is necessary to find the optimal parameters taking into account datapoints sampled from all classes. Since the absence of datapoints $\mathcal{X}^{u*}$ for unseen classes is a fundamental problem in ZSL, we therefore treat these missing datapoints as variables that should be *optimized* as well as model parameters. Usually, datapoints for training are sampled from a true distribution, and when generative model successfully approximates the target distribution, we can generate datapoints from the model randomly. Therefore, for the ideal case that the lower bound successfully catches the target distribution for both seen and unseen classes, the optimal parameters $\hat{\Theta}$ and *optimal* unseen datapoints $\mathcal{X}^{u*}$ should satisfy the following equations simultaneously:

$$\mathcal{X}^{u*}|_{\boldsymbol{A}_y^{u*}} \sim p\left(\boldsymbol{x}|\boldsymbol{a}_y^{u*}\right) = \int_{\boldsymbol{z}} p_{\hat{\theta}}\left(\boldsymbol{x}|\boldsymbol{z}\right) p_{\hat{\psi}}\left(\boldsymbol{z}|\boldsymbol{a}_y^{u*}\right) d\boldsymbol{z} \qquad (4)$$

$$\hat{\Theta} \simeq \underset{\Theta}{\operatorname{argmax}}\, \mathcal{L}\left(\Theta; \mathcal{X}^{s*}, \boldsymbol{A}_y^{s*}, \mathcal{X}^{u*}, \boldsymbol{A}_y^{u*}\right) \qquad (5)$$

As in (4), missing datapoints $\mathcal{X}^u$ can be *optimized* by sampling from the generative model, which optimally approximates the target distribution. This optimal generative model can be obtained with (5) by trained on that sampled datapoints $\mathcal{X}^{u*}$ of unseen classes, and existing datapoints of seen classes. Consequently, we can have a generative model which covers both seen and unseen classes by obtaining optimal parameters and sampled datapoints that satisfy (4) and (5).

In general, however, the optimal solution satisfying this chicken-egg problem is challenging to obtain in a closed form. To relax the problem, we can have the approximated solution by iteratively solving (4) and (5), namely Simultaneously Generating And Learning (SGAL) strategy. When collecting training data is possible such as the case of seen class, traditional training scheme for the optimal parameter of the model can be expressed as:

$$\hat{\Theta} = \underset{\Theta}{\operatorname{argmax}} \sum_{k=1}^{S} \frac{1}{N} \sum_{\boldsymbol{x}_n \sim p\left(\boldsymbol{x}|\boldsymbol{a}_k^{s*}\right)}^{N} \log p\left(\boldsymbol{x}_n|\boldsymbol{a}_k^{s*}; \Theta\right). \qquad (6)$$

However, collecting data from the target likelihood of unseen classes $p\left(\boldsymbol{x}|\boldsymbol{a}^{u*}\right)$ is impossible in this case. Instead, we can lean to the Expectaion-Maximization [4] by approximating the distribution of the auxiliary variable $\boldsymbol{x}^I$ which follows the graphical model shown in Fig. 1. In our case, $\boldsymbol{x}$ and $\boldsymbol{x}^I$ are assumed to be a feature vector and its corresponding image, respectively. Then EM formulation

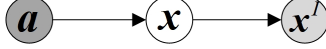

Figure 1: Graphical model for the EM formulation. The feature vector $\boldsymbol{x}$ is generated from the class attribute vector $\boldsymbol{a}$, and also generates the corresponding image $\boldsymbol{x}^I$. We assume that generating $\boldsymbol{x}$ is only affected by $\boldsymbol{a}$, and $\boldsymbol{x}^I$ is depend only on $\boldsymbol{x}$.

can be started with the following:

$$\log p\left(\boldsymbol{x}^I | \boldsymbol{a}^{u*}\right) = -\int_{\boldsymbol{x}} q\left(\boldsymbol{x}\right) \log \frac{p\left(\boldsymbol{x} | \boldsymbol{a}^{u*}; \Theta\right)}{q\left(\boldsymbol{x}\right)} d\boldsymbol{x} - \int_{\boldsymbol{x}} q\left(\boldsymbol{x}\right) \log \frac{q\left(\boldsymbol{x}\right)}{p\left(\boldsymbol{x}^I | \boldsymbol{x}\right) p\left(\boldsymbol{x} | \boldsymbol{a}^{u*}; \Theta\right)} d\boldsymbol{x}$$
$$= KL\left(q\left(\boldsymbol{x}\right) || p\left(\boldsymbol{x} | \boldsymbol{a}^{u*}; \Theta\right)\right) + \mathcal{L}\left(\Theta, q; \boldsymbol{a}^{u*}\right). \tag{7}$$

For Expectation step, we let $q\left(\boldsymbol{x}\right) = p\left(\boldsymbol{x} | \boldsymbol{a}^{u*}; \Theta^{old}\right)$ to let $KL$ term go to zero first. Note that $\Theta^{old}$ denotes the model parameter obtained in previous step. Substituting $q\left(\boldsymbol{x}\right)$ to (7) and maximizing $\mathcal{L}\left(\Theta, q\right) = \sum_{k=S+1}^{S+U} \mathcal{L}\left(\Theta, q; \boldsymbol{a}_k^{u*}\right)$ for the Maximization step, we have:

$$\underset{\Theta}{\operatorname{argmax}} \mathcal{L}\left(\Theta, q\right)$$

$$= \underset{\Theta}{\operatorname{argmax}} \sum_k -\int_{\boldsymbol{x}} p\left(\boldsymbol{x} | \boldsymbol{a}_k^{u*}; \Theta^{old}\right) \log \frac{p\left(\boldsymbol{x} | \boldsymbol{a}_k^{u*}; \Theta^{old}\right)}{p\left(\boldsymbol{x}^I | \boldsymbol{x}\right) p\left(\boldsymbol{x} | \boldsymbol{a}_k^{u*}; \Theta\right)} d\boldsymbol{x}$$

$$= \underset{\Theta}{\operatorname{argmax}} \sum_k \underbrace{-KL\left(p\left(\boldsymbol{x} | \boldsymbol{a}_k^{u*}; \Theta^{old}\right) || p\left(\boldsymbol{x}^I | \boldsymbol{x}\right)\right)}_{const} + \underset{\boldsymbol{x} \sim p\left(\boldsymbol{x} | \boldsymbol{a}_k^{u*}; \Theta^{old}\right)}{\mathbb{E}} \left[\log p\left(\boldsymbol{x} | \boldsymbol{a}_k^{u*}; \Theta\right)\right]$$

$$= \underset{\Theta}{\operatorname{argmax}} \sum_k \underset{\boldsymbol{x} \sim p\left(\boldsymbol{x} | \boldsymbol{a}_k^{u*}; \Theta^{old}\right)}{\mathbb{E}} \left[\log p\left(\boldsymbol{x} | \boldsymbol{a}_k^{u*}; \Theta\right)\right]$$

$$\simeq \underset{\Theta}{\operatorname{argmax}} \sum_{k=S+1}^{S+U} \frac{1}{N} \sum_{\substack{N \\ \boldsymbol{x}_n \sim p\left(\boldsymbol{x} | \boldsymbol{a}_k^{u*}; \Theta^{old}\right)}} \log p\left(\boldsymbol{x}_n | \boldsymbol{a}_k^{u*}; \Theta\right). \tag{8}$$

Note that $p\left(\boldsymbol{x}^I | \boldsymbol{x}\right)$ is independent to $\Theta$, as the relation between $\boldsymbol{x}^I$ and $\boldsymbol{x}$ is predetermined by the pre-trained network such as VGGNet or GoogLeNet; $\boldsymbol{x}^I$ does not join the actual training for the proposed method.

Compared to (6), last line of (8) can be seen as series of process that sampling data from previous model $p\left(\boldsymbol{x} | \boldsymbol{a}^{u*}; \Theta^{old}\right)$, and maximizing current log-likelihood $\log p\left(\boldsymbol{x} | \boldsymbol{a}^{u*}; \Theta\right)$ which can be achieved by training VAE with (2). In other words, we gradually update parameter $\Theta = (\theta, \phi, \psi)$, while simultaneously generate the datapoints $\mathcal{X}^u$ as training data, from the incomplete distributions represented by the decoder and prior network of previous step. See Algorithm 1 for a basic approach to approximate the generative model for both seen and unseen classes. Overview of the network structure and training process for our model is also displayed in Fig. 2(b). In the actual implementation of the proposed method, we initialize the model parameter $\Theta$ with converged network trained on labeled datapoints for seen classes, in order to ensure convergence and to exploit the seen classes as much as possible.

In (4), we assume that the model parameters are deterministic variables. However, unlike $\mathcal{X}^{s*}$ which is sampled from the true distribution, $\mathcal{X}^u$ is generated from the incomplete model which is still in the training process. In this case model uncertainty can take the place to disturb the datapoint generation. We thus handle the uncertainty and create datapoints in more general way, by assuming the model parameters to be Bayesian random variables. The conditional probability for unseen classes in (4) is approximately expressed as the following:

$$p\left(\boldsymbol{x} | \boldsymbol{a}^{u*}\right) = \int_{\theta, \phi, \boldsymbol{z}} p\left(\boldsymbol{x} | \boldsymbol{z}, \theta\right) p\left(\boldsymbol{z} | \boldsymbol{a}^{u*}, \psi\right) p\left(\theta\right) p\left(\psi\right) d\boldsymbol{z} d\theta d\psi$$

$$\simeq \sum_{l=1}^{L} \sum_{l'=1}^{L'} \int_{\boldsymbol{z}} p\left(\boldsymbol{x} | \boldsymbol{z}, \theta_l\right) p\left(\boldsymbol{z} | \boldsymbol{a}^{u*}, \psi_{l'}\right) d\boldsymbol{z} \tag{9}$$

where $\theta_l \sim p\left(\theta\right)$ and $\psi_{l'} \sim p\left(\psi\right)$. The prior distributions of parameters can be approximated with variational likelihoods, which are represented as Bernoulli distributions implemented with dropouts

**Algorithm 1** Simultaneously Generating-And-Learning Algorithm

**Require:** $\mathcal{X}^{s*}$, $\boldsymbol{A}_y^{s*}$ and $\boldsymbol{A}^{u*}$

1: $\Theta \leftarrow$ Initialize parameters with $\hat{\Theta} = \operatorname{argmax}_{\Theta} \mathcal{L} \left( \Theta; \mathcal{X}^{s*}, \boldsymbol{A}_y^{s*} \right)$
2: **while** $\Theta$ converges **do**
3: $\quad \mathcal{X}^{s_M^*}, \boldsymbol{A}_y^{s_M^*} \leftarrow$ Sample $M$ datapoints from $\mathcal{X}^{s*}$, $\boldsymbol{A}_y^{s*}$ as a minibatch
4: $\quad \boldsymbol{A}_y^{u_N^*} = \{\boldsymbol{a}_{y_n}^{u*}\}_{n=1}^N \leftarrow$ Randomly choose unseen class vectors from $\boldsymbol{A}^{u*}$ for $N$ times
5: $\quad \mathcal{X}^{u_N} = \{\boldsymbol{x}_n^u\}_{n=1}^N \leftarrow$ Sample $\boldsymbol{x}_n^u$ from $p\left(\boldsymbol{x}|\boldsymbol{a}_{y_n}^{u*}\right) \simeq \int p_\theta \left(\boldsymbol{x}|\boldsymbol{z}\right) p_\psi \left(\boldsymbol{z}|\boldsymbol{a}_{y_n}^{u*}\right) d\boldsymbol{z}$
6: $\quad \boldsymbol{g} \leftarrow \nabla_{\Theta}\mathcal{L}\left(\Theta; \mathcal{X}^{s_M^*}, \boldsymbol{A}_y^{s_M^*}, \mathcal{X}^{u_N}, \boldsymbol{A}_y^{u_N^*}\right)$
7: $\quad \Theta \leftarrow$ Update parameters using gradients $\boldsymbol{g}$ (e.g. Adam [11])
8: **end while**
9: **return** $\Theta$

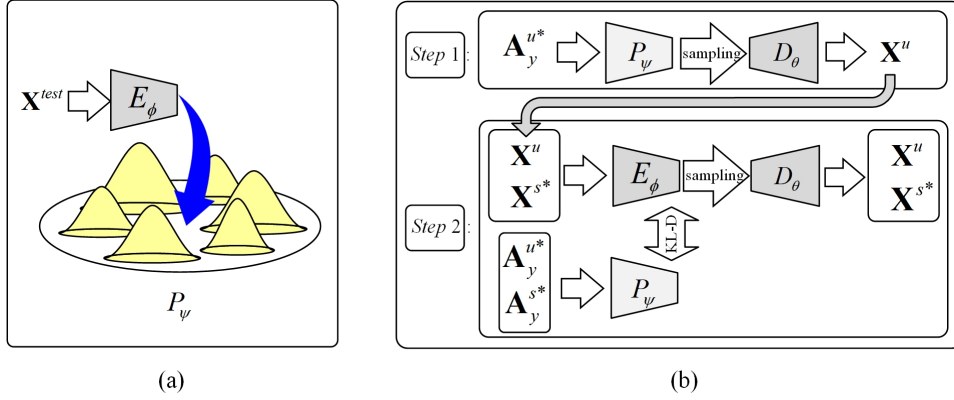

(a)　　　　　　　　　　　　　　　　　(b)

Figure 2: Overview of the proposed method. (a) Encoder as a classifier. Test datapoint is projected into latent space by encoder, where multi-modal prior exists represented by prior network. By calculating Euclidian distance between projected datapoint and multi-modal clusters, category is determined. (b) $P_\psi$, $D_\theta$ and $E_\phi$ denote the prior network for $p_\psi \left(\boldsymbol{z}|\boldsymbol{a}\right)$, decoder for $p_\theta \left(\boldsymbol{x}|\boldsymbol{z}\right)$ and encoder for $q_\psi \left(\boldsymbol{z}|\boldsymbol{x}\right)$ respectively. For training, we iteratively perform two steps. *Step* 1: Generating datapoints for unseen classes using current model, $p\left(\boldsymbol{x}|\boldsymbol{a}_y^{u*}\right) = \int_{\boldsymbol{z}} p_\theta \left(\boldsymbol{x}|\boldsymbol{z}\right) p_\psi \left(\boldsymbol{z}|\boldsymbol{a}_y^{u*}\right) d\boldsymbol{z}$. *Step* 2: Learning the model on both seen (existing training dataset) and unseen (generated dataset) classes using variational lower-bound.

[9]. Therefore, by activating dropouts when generating datapoints, parameter samplings expressed with summation in (9) can easily be achieved. In other words, while sampling datapoints of unseen classes using decoder $p_\theta \left(\boldsymbol{x}|\boldsymbol{z}\right)$ and prior network $p_\psi \left(\boldsymbol{z}|\boldsymbol{a}^{u*}\right)$, model uncertainty can be considered by activating dropouts in each network.

## 4 Experiments

### 4.1 Datasets and Settings

We firstly use the two benchmark datasets: AwA (Animals with Attributes) [16], which contains 30,745 images of 40/10(train/test) classes, and CUB (Caltech-UCSD Birds-200-2011) [26], comprised of 11,788 images of 150/50(train/test) species. Even though these benchmarks are selected by many existing ZSL approaches before [30], some unseen classes exist in the ImageNet 1K datasets. Since the ImageNet dataset is exploited to pre-train the various image embedding networks which are used as image-feature extractor for the datasets, these conventional setting breaks the assumption of zero-shot setting. We thus additionally choose 4 datasets [30] following the generalized ZSL (GZSL) setting, which guarantees that none of the unseen classes appear in the ImageNet benchmark: AwA1, AwA2, CUB and SUN. AwA1 and AwA are the same dataset but AwA1 is rearranged to follow the GZSL setting. AwA2 is an extension version of AwA and contains 37,322 images of 40/10(train/test) classes. SUN is a scene-image dataset and consists of 14,340 images with 645/72/65(train/test/validation)

Table 1: Comparision of the zero-shot classification accuracy (%) on AwA and CUB with conventional setting. F: how the image feature vector is obtained for non neural network approaches. $F_G$ for GoogLeNet and $F_V$ for VGGNet. For deep models, $N_G$ for Inception-V2(GoogLeNet with batch-normalization), and $N_V$ for VGGNet. SS : semantic space. A: attribute space. W:semantic word vector space. mmVAE and SGAL denote our models trained as normal multi-modal VAE with seen classes and trained in generating-and-learning manner, respectively.

| Methods | F | SS | AwA 10-way 0-shot | CUB 50-way 0-shot |
|---------|---|----|-----------------|-----------------|
| SJE[2] | $F_G$ | A | 66.7 | 50.1 |
| ESZSL[21] | $F_G$ | A | 76.3 | 47.2 |
| SSE-RELU[35] | $F_V$ | A | 76.3 | 30.4 |
| JLSE[36] | $F_V$ | A | 80.5 | 42.1 |
| SYNC-STRUCT[6] | $F_G$ | A | 72.9 | 54.5 |
| SEC-ML[5] | $F_V$ | A | 77.3 | 43.3 |
| DEVISE[8] | $N_G$ | A/W | 56.7/50.4 | 33.5 |
| SOCHER *et al.*[22] | $N_G$ | A/W | 60.8/50.3 | 39.6 |
| MTMDL[31] | $N_G$ | A/W | 63.7/55.3 | 32.3 |
| BA *et al.*[17] | $N_G$ | A/W | 69.3/58.7 | 34.0 |
| SAE[14] | $N_G$ | A | 84.7 | 61.4 |
| DEM[34] | $N_G$ | A/W | **86.7**/78.8 | 58.3 |
| RELATIONNET[24] | $N_G$ | A | 84.5 | 62.0 |
| VZSL[28] | $N_V$ | A | 85.3 | 57.4 |
| mmVAE | $N_G$ | A | 74.2 | 58.4 |
| SGAL | $N_G$ | A | 84.1 | **62.5** |

classes. These datasets under GZSL setting, are more suitable to the realistic zero-shot problems in practice.

## 4.2 Network Structure and Training

Similar to the previous works [17, 20, 24, 34], we use image embedding networks for ZSL. For the conventional setting, Inception-V2 [25] is used and ResNet101 [10] for the GZSL setting. Since the proposed method exploits VAE with multi-modal latent prior, our network structure is composed of encoder, decoder and prior network as shown in Fig. 2(b). All parts of our model are basically constructed with dense (or fully connected) layers. For computational complexity and memory requirements, network structure and parameters can be a standard to examine the complexity and memory requirements, and we compare ours with other generative-based methods: for ours on AwA2, 1 hidden layer with 512 units is used for both encoder and decoder. In [15], 2 and 1 with both 512 units are used for encoder and decoder, respectively. In [18], 2 with 512 and 1 with 1024 are used for encoder and decoder respectively. [29] uses 1 with 4096 for generator, and 1 with 1024 for discriminator. We will add this evaluation to our paper. Details of the network structures and parameter settings can be found in our supplementary.

Before applying the proposed SGAL strategy, we first pre-train our model on the seen classes, as shown in Algorithm 1. We found that learning diverges when training is proceeded for both seen and unseen classes from the beginning. Once the pre-training converges, we perform fine-tuning for both seen and unseen classes subsequently; iteratively sampling and learning the minibatch by generating datapoints for unseen classes. The number of iterations for the benchmarks are: for mmVAE and SGAL(EM), 170,000 and 1,300 for AwA1, 64,000 and 900 for AwA2, 17,000 and 2,000 for CUB1 and 1,450,000 and 1,500 for SUN1. In order to consider the model uncertainty, we also train the model adopting (9) when generating unseen datapoints. For one latent variables sampled from prior network, a total of 5 samples are generated while activating dropouts in the decoder. Unlike (9), in the actual implementation all the dropouts of the prior network are deactivated for the training stabilization.

Table 2: Zero-shot classification comparison results with GZSL setting. Methods are evaluated using Top-1 accuracy (%) on **u**: unseen classes, **s**: seen classes. **H**: Harmonic mean of **u** and **s** is also reported. mmVAE, SGAL and SGAL-dropout denote our models trained as plane multi-modal VAE with seen classes, trained in generating-and-learning manner for both seen and unseen classes, trained with activated dropouts when generating unseen datapoints, respectively.

| | AwA1 | | | AwA2 | | | CUB | | | SUN | | |
|---|---|---|---|---|---|---|---|---|---|---|---|---|
| Methods | u | s | H | u | s | H | u | s | H | u | s | H |
| CONSE[19] | 0.4 | 88.6 | 0.8 | 0.5 | 90.6 | 1.0 | 1.6 | 72.2 | 3.1 | 6.8 | 39.9 | 11.6 |
| DEVISE[8] | 13.4 | 68.7 | 22.4 | 17.1 | 74.7 | 27.8 | 23.8 | 53.0 | 32.8 | 16.9 | 27.4 | 20.9 |
| ESZSL[21] | 6.6 | 75.6 | 12.1 | 5.9 | 77.8 | 11.0 | 12.6 | 63.8 | 21.0 | 11.0 | 27.9 | 15.8 |
| ALE[1] | 16.8 | 76.1 | 27.5 | 14.0 | 81.8 | 23.9 | 23.7 | 62.8 | 34.4 | 21.8 | 33.1 | 26.3 |
| SYNC[6] | 8.9 | 87.3 | 16.2 | 10.0 | 90.5 | 18.0 | 11.5 | 70.9 | 19.8 | 7.9 | 43.3 | 13.4 |
| SAE[14] | 1.8 | 77.1 | 3.5 | 1.1 | 82.2 | 2.2 | 7.8 | 57.9 | 29.2 | 8.8 | 18.0 | 11.8 |
| DEM[34] | 32.8 | 84.7 | 47.3 | 30.5 | 86.4 | 45.1 | 19.6 | 54.0 | 13.6 | 20.5 | 34.3 | 25.6 |
| RELATION[24] | 31.4 | **91.3** | 46.7 | 30.0 | **93.4** | 45.3 | 38.1 | 61.1 | 47.0 | - | - | - |
| SRZSL[3] | - | - | - | 20.7 | 73.8 | 32.3 | 24.6 | 54.3 | 33.9 | 20.8 | 37.2 | 26.7 |
| CVAE-ZSL[18] | - | - | 47.2 | - | - | 51.2 | - | - | 34.5 | - | - | 26.7 |
| f-CLSWGAN[29] | - | - | - | 57.9 | 61.4 | 59.6 | 43.7 | 57.7 | **49.7** | 42.6 | 36.6 | **39.4** |
| SE-GZSL[15] | **56.3** | 67.8 | 61.5 | **58.3** | 68.1 | 62.8 | 41.5 | 53.3 | 46.7 | 40.9 | 30.5 | 34.9 |
| mmVAE | 39.4 | 86.8 | 54.2 | 15.7 | 92.6 | 26.9 | 28.5 | 63.1 | 39.3 | 14.2 | **43.6** | 21.4 |
| SGAL | 52.7 | 74.0 | 61.5 | 52.5 | 86.3 | 65.3 | 40.9 | 55.3 | 47.0 | 35.5 | 34.4 | 34.9 |
| SGAL-dropout | 52.7 | 75.7 | **62.2** | 55.1 | 81.2 | **65.6** | **47.1** | 44.7 | 45.9 | **42.9** | 31.2 | 36.1 |

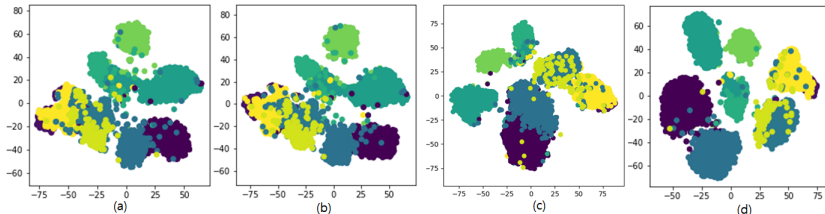

Figure 3: Structure visualization of learned dataset AwA1,2. Each color denotes unseen classes. Results of (a) mmVAE on AwA1, (b) SGAL on AwA1, (c) mmVAE on AwA2 and (d) SGAL on AwA2. While harmonic mean score is increased from 52.2% to 62.2% on AwA1, there are less drastic changes between (a) and (b). On the other hand, increased from 26.9% to 65.6% on AwA2, clusters are more separated from each other in (d) compared to (c).

## 4.3 Evaluation Results with Conventional and GZSL Settings

To evaluate the proposed method, we first compare several alternative approaches on the conventional setting, and display the results in Table 1. Note that in most works for ZSL with conventional setting, it is assumed that the test data only comes from the unseen classes. Our method obtains competitive result when evaluated on AwA, and state-of-the-art performance on more challenging CUB benchmark dataset. We also test our method on GZSL setting under the disjoint assumption as proposed by [30]. As a measure of performance for this generalized setting, we obtain classification accuracy for both seen and unseen classes, and report the harmonic mean of the two accuracies. Results are shown in Table 2. Our model outperforms than other non-generative methods, and shows competitive results compared to the models based on generative models [28, 18, 29, 15]. Note that other generative-based methods mainly use additional off-the-shell classifier, after generating estimated samples of unseen classes with their model. In our case, however, the encoder serves as a classifier since the proposed model covers seen and unseen classes by itself.

## 4.4 Effects of Generating And Learning, and Dropout Activation

The proposed approach is based on the VAE with multi-modal prior trained on seen classes, and learns the unseen classes through SGAL strategy. Additionally, model uncertainty can be handled by dropouts while generating the missing datapoints for unseen classes. This series of steps can be applied in order, and we show the evaluation results with each step's model in the bottom two rows in Table 1, and bottom three rows in Table 2: mmVAE indicates the VAE with multi-modal prior trained only on the seen classes as in Section 3.2, SGAL is for the model with SGAL strategy, and SGAL-dropout denotes the SGAL model activating dropouts in the decoder when generating unseen datapoints. In the case of mmVAE, it shows low performance for unseen classes since the model learns the target distribution of only the seen classes. However, SGAL generates missing datapoints by using class embeddings and the model itself, and the entire model is trained from that generated datapoints and seen class datapoints iteratively. As SGAL aims to learn the distributions of both seen and unseen classes in this manner, robust classification performance of unseen classes is achieved. One can observe that SGAL shows the decreased performance for the seen classes rather than mmVAE. We believe that the proposed method is a generative model that covers the distribution for all classes, thus the performance trade-off between seen and unseen classes occurs. In order to visualize the effects of the proposed method, several learned datasets are displayed in Fig. 3 using T-SNE.

The proposed model shows state-of-the-art results in harmonic mean on AwA1 and AwA2 dataset, and in classification accuracy of unseen on CUB and SUN dataset; CUB and SUN datasets contain almost 5 and 12 times more classes than AwAs, and the multi-modal distribution for seen classes is distorted more easily when fine-tuning for inserting new clusters for unseen. That is, unseen clusters can be deduced based on plenty of seen clusters thus the model achieves outperformed results for unseen, but performance drops more easily for seen classes due to the distortion.

In general, the generative model leans to the training dataset sampled from the real-world, but in SGAL strategy the model learns the target distribution from the datapoints sampled from the distribution which the model itself represents. Since the generated datapoints float depending on the current model, the model uncertainty can affect the model performance. To relieve the problem, SGAL-dropout uses dropout activation when sampling unseen datapoints and shows more robust classification results compared to that of SGAL's. That is, by sampling the unseen datapoints while reducing the model uncertainty, the model better describes the target distribution of unseen classes. In this case, however, the performance for the seen classes is further reduced by the generalization for both seen and unseen classes, similar to the case between mmVAE and SGAL.

## 5   Conclusion

We have introduced a novel strategy for zero-shot learning (ZSL) using VAE with multi-modal prior distribution. Absence of the datapoints for unseen classes is the fundamental problem of ZSL, which makes it challenging to obtain a generative model for both seen and unseen classes. We therefore treat the missing datapoints as variables that should be *optimized* like model parameters, and train our network with Simultaneously Generating-And-Learning strategy similar to EM manner. In other words, while training our model iteratively generate unseen samples and use them as training datapoints to gradually update model parameters. Consequently, our model favorably attain both seen and unseen classes understanding. With the encoder and the prior network, classification can be performed directly without additional classifiers. Further, by catching the model uncertainty with dropouts, we show that a more robust model for unseen classes is achievable. The proposed method has competitive results with the state-of-the arts on various benchmarks, while outperforming them for several datasets.

**Acknowledgments**

We would like to thank Jihoon Moon and Hanjun Kim, who give us intuitive advices. This work was supported by the National Research Foundation of Korea(NRF) grant funded by the Korea government(MSIP) (No. 2017R1A2B2002608), in part by Automation and Systems Research Institute (ASRI), and in part by the Brain Korea 21 Plus Project.

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
