[Supplementary Material]

# Supplementary for
# Zero-shot Learning via Simultaneous Generating and Learning

## 1  Introduction

In this document, we present detailed network structures and hyper parameters used for training of each dataset. Our model is based on VAE with multi-modal prior distribution. To achieve distinct multi-modal for clear clustering in latent space, we restrict the prior distribution with regularization loss $L_{reg}$ as [2]; clusters for each category are enforced to be separated from each other within certain distance, $D_z \times d$, where $D_z$ is the dimension of the latent variable, and $d$ is a threshold which is chosen manually. The total loss for our model is then expressed as the following:

$$L_{total} = \mathcal{L}\left(\Theta; \mathcal{X}^*, \boldsymbol{A}_y^*\right) + \gamma L_{reg}\left(D_z, d\right), \tag{1}$$

where $\mathcal{L}$ is the lower bound of VAE, and $\gamma$ is a hyperparameter for balanced training procedure.

In both conventional and GZSL settings, datasets vary in the number of total categories and the datapoint dimension, and our approach is to place distinct Gaussian distributions corresponding to classes in the latent space. For this reason, it is necessary to adjust the network structure and hyperparameters according to the dataset to secure the capacity to compress all the class distributions into the latent space. Hence we empirically determine proper network structures, and choose $d$ and $\gamma$. For the encoder, decoder and the prior network we use maximum 3, 4 and 6 blocks respectively, with slight variations depending on the datasets. To form a unit block for networks, we use leaky-relu activation function, batch normalization and dropout layers sequentially after a dense layer. For the last blocks of encoder, decoder and prior network, we omit the activation, batch normalization and dropout.

For ZSL, we first pre-train our model on seen classes, and perform fine-tunning for both seen and unseen classes with SGAL strategy. Before fine-tuning, the model is highly-fitted to the seen classes; we observe that under this condition our model hardly learn the unseen classes, when the ratio of unseen and seen in one minibatch is equal to the ratio related number of classes, or simply 0.5. We empirically found that our model achieves the best performance when $g$, the ratio of unseen, is between 0.8 and 0.95. Training process is terminated when the maximum classification accuracy of unseen classes, or maximum harmonic mean is achieved for conventional and GZSL settings respectively. In order to prevent the network from diverging, we also train our model on only seen classes with probability $p$, while SGAL training is iteratively conducted. We specify the parameter values and network settings for each testbed in Table. 1. For the optimizer, Adam [1] is used for all cases with learning rate $10^{-4}$.

Table 1: Hyperparameters and network structures for each dataset. To make the training procedure be stabilized, we weight the regularization loss $L_{reg}$ with $\gamma$. $d$ is the distance threshold for the separation of multi-modal in latent space. For SGAL strategy, we crate minibatch composed of both seen and unseen classes, and $g$ denotes the ratio of unseen one. Since the proposed learning has the performance trade-off between accuracies of seen and unseen classes, we also train our model only on seen classes with probability $p$, while training with SGAL strategy is iteratively performed. The same dropout rates of the decoder are used for both training time and generating fake samples when adopting SGAL-dropout method.

| | | conventional ZSL | | GZSL | | | |
| --- | --- | --- | --- | --- | --- | --- | --- |
| | | AwA | CUB | AwA1 | AwA2 | CUB | SUN |
| Encoder | input dim | 1024 | 1024 | 2048 | 2048 | 2048 | 2048 |
| | layers | 3(#)<br>(512, 512, 512x2) | 3(#)<br>(1024, 1024, 256x2) | 2(#)<br>(512, 64x2) | 2(#)<br>(512, 256x2) | 2(#)<br>(512, 64x2) | 2(#)<br>(2048, 128x2) |
| | dropout rate | 0.5 | 0.2 | 0.5 | 0.5 | 0.5 | 0.5 |
| Decoder | input dim | 512 | 256 | 64 | 64 | 64 | 128 |
| | layers | 4(#)<br>(512, 512, 512, 1024) | 4(#)<br>(1024, 1024, 1024, 1024) | 1(#)<br>(2048) | 2(#)<br>(512, 2048) | 2(#)<br>(512, 2048) | 3(#)<br>(2048, 2048, 2048) |
| | dropout rate | 0.5 | 0.2 | 0.5 | 0.5 | 0.5 | 0.5 |
| Proirnet | input dim | 85 | 312 | 85 | 85 | 312 | 102 |
| | layers | 2(#)<br>(…,512) | 6(#)<br>(…,256) | 6(#)<br>(…,64) | 6(#)<br>(…,64) | 6(#)<br>(…,64) | 3(#)<br>(…,128) |
| | dropout rate | 0.2 | 0.2 | 0.2 | 0.2 | 0.5 | 0.2 |
| $\gamma$ | | 0.01 | 0.01 | 0.1 | 0.1 | 0.01 | 0.01 |
| $d$ | | 1.2 | 1.2 | 2.0 | 2.0 | 1.2 | 1.2 |
| batch size | | 64 | 128 | 128 | 128 | 128 | 128 |
| $g$ | | 0.9 | 0.9 | 0.9 | 0.95 | 0.85 | 0.95 |
| $p$ | | - | - | 0.5 | 0.3 | 0.5 | 0.1 |