[Reviews · NeurIPS 2019]

Reviewer 1



Overview: The authors propose an original approach to zero-shot-learning by combining VAEs with EM for inferring the optimal unseen examples. The key idea is simultaneously generating examples of unseen classes and learning from them. The authors run a number of experiments which demonstrate that the proposed method shows competitive performance in a number of ZSL tasks. Quality: The work is generally of high quality. The experiments are clearly described, and the model specifications are detailed. What limits its quality, however, is the lack of performance variability estimates. In general, the results are not entirely consistent across metrics, so it seems that adding variability measures is warranted. I also feel that in the paper may greatly benefit from providing a few toy simulation studies in low-dimensional settings, s.t. the structure of learned classes may be easily visualized. Clarity: The paper is generally well written and is a pleasure to read. It would help to clarify how the class embeddings are constructed (even if they are given in the current setting) There is also a number of minor typos: 234: outperforms than, 277 attain(s) Originality: The proposed method is certainly novel and the contribution is original enough. Conclusion: Overall, the paper proposes a novel approach to zero-shot learning. In my view, it’s a borderline submission. Overall, while there are considerable limitations, I think that this paper is slightly above the acceptance threshold. ## After reading the authors' response. The authors partially addressed some of my concerns, providing variability measures for some of the results and an additional low-dimensional visualization. I believe that to further elucidate the behavior of the algorithm, it could still be beneficial to actually train on an artificial low-dimensional dataset, as opposed to using T-SNE visualization on a natural dataset. At the same time, I partially agree with some of the concerns voiced by the 2nd reviewer in that more experimentation would be helpful to understand the contributions of different parts of the model. This is partially why I think low-dimensional simulations would be helpful. Overall, I still think that this is a borderline submission, slightly above the threshold and I keep my score unchanged.

Reviewer 2



Comments -------- notation comment -- it would be great to simplify the readability of the notations, particularly the 'asterix' was hard for me to read. Here is one suggestion that I have, please consider adapting it, {x_i, y_i} as the labeled set, y_i \in {Y}, and {u_i, z_i} as unlabeled set z_i \in {Z} and Y \intersect Z is null. motivation of why to use VAE -- "In order to capture the complex distribution, VAE can be a useful tool." seems a bit weak, there are a few concrete choices here, VAE vs specificying a parametric generative distribution vs GAN. Given the experiments on image dataset, why VAE ? A key difference from prior work is the reliance of the number of unseen classes and their attributes to be known apriori, I wonder how realistic this assumption is, most other work rely on a new attribute (a.k.a class) being provided at inference time, whereas this work cannot handle that scenario. A number of key ablations seem to be missing, (a) How much is the EM actually helping ? (b) how much is the *multi-modal* prior actually helping ? (c) how big are the model parameters, are they comparable to other work ? (d) if the number of EM steps is just 1, this is similar to some of the other cited work. Specifically, when the parameters are optimized based on seen classes fully and the unseen classes attributes are used to generate examples on which a deep-NN classifier is trained. How much of the benefit comes from having a more expressive deep-NN vs a simpler baseline like AVM ? What if we made some work (there are many, just to cite one from the paper - 'Generalized Zero-Shot Learning via Synthesized Examples') use a deep-NN than the svm used by the cited work ? (e) How much of the benefit even just from this work can be improved by using a discriminative deep-NN instead of a generative one ? Main concern [Originalty & Significance] ---------------------------------------- Significance: Most of this work to me seemed to me like "yes, I think if you do that it is likely performance will improve compared to other work", I found the work to be more like a bag of tricks than some key insight that the work was providing. While overall this is not a bad thing and there can be good work that improve the sota on well-established benchmarks, this particular work worries me in tending to just improve end-2-end scores without providing any key-insights. Originality: AFAIU, the new insights seem to be using EM, and multimodal, both of which are relatively mild. Focusing on running ablations will perhaps help the authors to focus on the key contributions and refine their work. After rebuttal ------------------ Thank you for the responses and the references, they were helpful. They have addressed my questions/concerns (partially). Overall, I think there is still room to perform clearer ablations to study the effect of each component, for e.g. one proposal/contribution here is the generation of samples for unseen classes.. for a practitioner it would be great to have some clear set of experiments proving its utility (as a thought, the generated samples can be used to train a discriminative classifier than continuing to be a generative one.)

Reviewer 3



The article is indicative of solid work and it is well written. The state of the art is covered well. The evaluation does not elaborate on the computational complexity and memory requirements of the proposed algorithm compared to the state of the art systems. It is not fully clear how the Simultaneously Generating And Learning (SGAL) strategy generates the missing datapoints by using class embeddings and the model itself. Elaborating this mechanism would help the reader. The multi-modal prior plays a crucial role in providing information for supporting the learning of the unseen cases. Please elaborate the number of iterations that the algorithm requires for the benchmark datasets. Reviewer comment to author response: the authors have provided more details on the above points and addressed the mentioned presentation related issues.

[Author Response · NeurIPS 2019]

**Reviewer#1 - 1) Adding variability measures for the results** As recommended, we will add variability measures, for
example, harmonic mean (average of seen and unseen), $39.9 \pm 11.3\%$([18]), $49.6 \pm 10.1\%$([29]), $51.5 \pm 13.2\%$([15])
and $52.4 \pm 13.9\%$(ours). **2) Visualizing the learned classes with low-dimensional toy simulation studies** We notice
that the reviewer encouraged us to provide some low-dimensional toy simulations which could help other readers easily
understanding what is going on. Since our network encodes, for example in AwA dataset, 4096D datapoints into 64D
latent variables, we could display the structure of several datasets directly using T-SNE as shown in Fig. 1, especially
for unseen classes. We hope it will be better supplements than recommended, then we will include this figure.

Figure 1: Structure visualization of learned dataset AwA1,2. Each color denotes unseen classes. Results of (a) mmVAE
on AwA1, (b) SGAL on AwA1, (c) mmVAE on AwA2 and (d) SGAL on AwA2. While harmonic mean score is
increased from 52.2% to 62.2% on AwA1, there are less drastic changes between (a) and (b). On the other hand,
increased from 26.9% to 65.6% on AwA2, clusters are more separated from each other in (d) compared to (c).

**Reviewer#2 - 1) Notation** As recommended, we will reexamine the notations and try to modify them to simple
forms. **2) Why VAE?** To generate datapoints and perform feedback training to catch intractable distributions, we
need encoder-decoder structured non-parametric generative model like VAE. Although GAN is also a powerful model,
absence of encoder limits our intend to implement feedback training and regularization of encoded latent variables
and multi-modal prior distribution. **3) Reliance of the unseen classes** Previous works such as [8,19,21] aim to have
semantic embedding model to cope with unknown attributes. On the other hand, [3,15,18,29] exploit generalization of
generative models for zero-shot problems, assuming known attributes in order to generate samples for unseen classes
on which classifiers are trained. Therefore, in our opinion, unknown attributes would be the better assumption for
zero-shot problems but it is still worth studying with known attributes similar to other works mentioned above. **(a-e)**
Please note that we aim to train a generative model for both seen and unseen classes, and overcome lack of training
data for unseen by approximating missing samples. In other words, in one iteration for training, the model generates
missing samples and is trained on both missing samples for unseen and existing ones for seen. This one iteration is
formulated by EM, thus **(d)** one EM step is equivalent to train our network with just one iteration for unseen which will
be insufficient to converge. **(a)** In order to examine EM, we show the results of mmVAE which is trained only on seen
classes without EM. And we do not have any deep-NN classifier, since we exploit encoded features for classification,
which is our other contribution. Specifically, **(b,d)** when our model is completely trained, we encode datapoints to
latent variables and determine their classes by calculating Euclidean distances to each multi-modal and choosing the
modal with minimum distance as Eqn. (3). **(d)** Even if the cited work [15] uses a deep-NN rather than the SVM, or if
we use additional classifier, still our purpose is different from others since we aim to have the model with both seen
and unseen; [15] also performs feedback with generated unseen datapoints, but only for updating decoder in order to
restrict their encoded latent variables partially. **(c)** In [15], for example, they use 2 and 1 hidden layers for encoder and
decoder with 512 hidden units, while ours use 1 and 1 with 512 for AwA dataset. **(e)** To perform generation and MLE,
discriminative deep-NN could hardly be adopted, but generative model could be.

**Reviewer#3 - 1) It is not fully clear how the SGAL strategy generates the missing datapoints** As the reviewer
commented, the multi-modal prior does play a crucial role. To generate missing datapoints by implementing Eqn. (4),
1) multi-modal prior generates the latent variables of each unseen class, and 2) decoder predicts the missing datapoints
by decoding these latent variables. Subsequently the whole model is trained on both the generated datapoints of unseen
and existing ones of seen. We will add this additional explanation in our paper. **2) The number of iterations for the
benchmarks** For mmVAE and SGAL(EM): 170,000 and 1,300 for AwA1, 64,000 and 900 for AwA2, 17,000 and 2,000
for CUB1 and 1,450,000 and 1,500 for SUN1. We will add this with a table in our paper. **3) Fake samples?** As the
reviewer suggested, approximations would be the better expression compared to fake samples, therefore we will modify
it as recommended. **4) Computational complexity and memory requirements** Network structure and parameters can
be a standard to examine the complexity and memory requirements, and we compare ours with other generative-based
methods: For ours on AwA2, 1 hidden layer with 512 units is used for both encoder and decoder. In [15], 2 and 1 with
both 512 units are used for encoder and decoder, respectively. In [18], 2 with 512 and 1 with 1024 are used for encoder
and decoder respectively. [29] uses 1 with 4096 for generator, and 1 with 1024 for discriminator. We will add this
evaluation to our paper.

[Meta-Review · NeurIPS 2019]

The addresses zero-shot learning by an EM process of iteratively generating examples from unseen classes, and learning with them, this leads to generating samples that are good to learn from. Reviewers found the idea novel for this context, the writing clear and the experiments (mostly) convincing. They asked to see additional ablation studies in the final version.